# Hot-Deformation Behavior and Microstructure Evolution of the Dual-Scale SiCp/A356 Composites Based on Optimal Hot-Processing Parameters

**DOI:** 10.3390/ma13122825

**Published:** 2020-06-23

**Authors:** Yahu Song, Aiqin Wang, Douqin Ma, Jingpei Xie, Zhen Wang, Pei Liu

**Affiliations:** 1School of Material Science and Engineering, Henan University of Science and Technology, Luoyang 471023, China; syh198703@163.com (Y.S.); jingpeixie@163.com (J.X.); hkdwangz@163.com (Z.W.); liupei_hkd@163.com (P.L.); 2CITIC Heavy Industries CO., Ltd., Luoyang 471039, China; 3Provincial and Ministerial Co-construction of Collaborative Innovation Center for Non-ferrous Metal New Materials and Advanced Processing Technology, Luoyang 471023, China; madouqin1987@163.com

**Keywords:** dual-scale SiCp/A356 composites, hot deformation behavior, microstructure, constitutive equation, processing map

## Abstract

Hot deformation at elevated temperature is essential to densify particle-reinforced aluminum matrix composites (AMCs) and improve their performance. However, hot deformation behavior of the AMCs is sensitive to the variation of hot-processing parameters. In this paper, optimal processing parameters of dual-scale SiCp/A356 composites were determined to explore the control strategy of the microstructure. Hot-compression tests were conducted at temperatures ranging from 460 to 520 °C under strain rates from 0.01 to 5 s^−1^. Constitutive equation and processing maps were presented to determine the hot-processing parameters. Microstructure evolution of the dual-scale SiCp/A356 composites was analyzed. The strain rate of 0.62–5 s^−1^ and deformation temperature of 495–518 °C is suitable for the hot processing. The number of dynamic recrystallization (DRX) grains in the “safe” domains is larger and the dislocation density is lower compared to those of instability domains. DRX grains mainly occurred around SiC particles. The presence of SiC particles can promote effectively the DRX nucleation, which results in the dynamic softening mechanism of the dual-scale SiCp/A356 composites being dominated by DRX.

## 1. Introduction

Due to their low density and expansion coefficient, high specific strength and specific modulus, good high-temperature performance and wear resistance, SiC particle-reinforced Al–Si matrix (A356, A357) materials are appropriate for weight reduction in the field of the automotive industry and other engineering applications [1,2,3,4]. Traditionally, the SiCp/Al–Si matrix composite is usually fabricated using either micro-sized or nano-sized SiC particles. The properties of SiCp/Al–Si composites depend on the amount and size of SiC particles. When the SiC particles amount is determined, the micro-sized SiC particles could improve the hardness and strength of Al–Si matrix composites at the expense of plasticity. While nano-sized SiC particles could improve the plasticity of Al–Si matrix composites, the enhancement of strength and wear resistance is not remarkable. In recent years, it has been found that the simultaneous introduction of micro-and nano-sized particles into the Al–Si alloy could significantly improve the comprehensive properties with the idea of micro-sized SiC particles helping in enhancing the hardness, strength and wear resistance, while the nano-sized SiC particles help in retaining the plasticity [5,6,7].

It is well known that metal matrix composites often need secondary thermo-mechanical processing such as hot forging, hot extrusion and hot rolling before they are used as structural or functional material in industry, because the secondary thermo-mechanical processing could effectively regulate the mechanical properties and microstructure of composites [8,9,10]. In this case, the hot deformation behavior of composites should be strictly controlled during the secondary thermo-mechanical processing due to its effect on the microstructure and workability of composites. In recent years, researches have mainly focused on the hot deformation behavior of either micro-sized or nano-sized SiCp/Al composites [11,12,13,14,15,16,17,18], and the results showed that the deformation behavior of SiCp/Al composites is greatly influenced by the SiC particle condition (particle size, distribution and volume).

Dynamic recovery (DRV) and dynamic recrystallization (DRX) are the main softening mechanisms. DRV is prone to occur in high stacking-fault energy metals, because of the easy movement of dislocations. While DRX easily occurs in the low stacking-fault energy materials. For materials with low fault energy, the dislocation is not easy to move. When the dislocation accumulates to a certain extent, recrystallization is likely to occur to reduce the energy of the system. Dislocation density decreased significantly after DRX. At this time, dislocation could be actuated in recrystallized grains, so the stress decreased. Generally, DRX rarely occurs in aluminum alloy and its composites because of the high stacking fault energy of these materials [19]. In fact, DRX occurrs during hot deformation of aluminum alloys [8,18]. It is related to the size and distribution of SiC particles [20,21]. The particle deformation zones (PDZ) containing high dislocation density provide ideal sites for the development of a recrystallization nucleus and leadto local DRX. It should be mentioned that the size of hard particles, the inter-particle spacing and their morphologies can affect the conditions for the occurrence of DRX. Irregular SiC particles with a large size are prone to breakage during deformation, which leads to a rotation of the broken particles and a weak interface between the particles and the Al matrix, and relaxes the stress concentration; while the presence of nanoparticles leads to typical strain hardening due to their pinning effect on dislocations. Research of hot deformation behavior of particle-reinforced aluminum matrix composites (AMCs) is mainly focused on single micron or nano reinforced particles [17,22,23]. DRX is an important phenomenon for controlling microstructure and mechanical properties in hot working. Whether DRX occurs in the dual-scale SiCp/A356 composites need to be studied.

Based on the above analysis, hot-deformation behavior of the dual-scale SiCp/A356 composites was clarified using the hot compression flow curves in the present work. Constitutive equation and processing maps were presented to determine the hot-working constants. Microstructure evolution of the dual-scale SiCp/A356 composites was analyzed. Furthermore, hot deformation behavior was analyzed by the dislocation particulate interactions model.

## 2. Materials and Methods

The dual-scale SiCp/A356 composite was prepared via powder metallurgy (PM) by mixing micro-sized SiC powder with the average size of 10 μm, nano-sized SiC powder with the average size of 80 nm, and A356 alloy powder with the average size of 7 μm. The content of micro- and nano-sized SiC particles in the composite was 2 and 13 vol.%, respectively. The elemental compositions of the original A356 alloy powder are shown in Table 1.

The preparation process of the composite was as follows: Firstly, the nano-sized SiC particles were mixed with A356 aluminum alloy powder by the planetary ball mill (QM-BP type, Zhuo De Instrument (Shanghai) Co., LTD, Shanghai, China) at a rotation speed of 150 r/min for 12 h, then the micro-sized SiC particles were mixed with the composite powders with the same process. Secondly, the mixed powders were pressed by using a 500 T four-column hydraulic press at 500 MPa for 60 min, and then the cold-pressed samples were sintered in an argon atmosphere-protected sintering furnace (SG-GL1200 type, Shanghai Jujing Precision Instrument Manufacturing Co. LTD, Shanghai, China) at 550 °C for 4 h. After that, the sintered samples were hot extruded using a horizontal extruder (XJ-500 type, Wuxi Jishun Air Separation Equipment Co. LTD, Wuxi, China) at 500 °C with an extrusion ratio of 15:1 and extrusion speed of 1 mm/s. Finally, the composite was annealed in the gas-protected sintering furnace at 300 °C for 2 h. The microstructure of composites was analyzed with an optical microscope (OM, Axiovert 200MAT, Carl Zeiss AG, Jena, Germany) and transmission electrical microscope (TEM, JEM-2100, Japan Electronics Co. LTD, Tokyo, Japan). The dual-scale SiCp/A356 composites comprise black micro SiC particles and gray Si phase (Figure 1a). The nano SiC particles were further examined in detail by TEM (Figure 1b). Nano SiC particles are uniform distributed on the Al matrix.

Finally, the composite was processed into small cylindrical samples with the size of Φ8 mm × 12 mm by wire cutting, and a small hole of Φ0.5 mm × 2.5 mm was machined on the side of the samples to install the thermocouple. Hot-compression tests were taken by Gleeble-1500D thermal simulator (Data Sciences International, INC., St. Paul, MN, USA) at the temperatures of 460, 480, 500, 520 °C and the strain rates of 0.01, 0.1, 1, 5 s^−1^, respectively. True strain value was set as 0.7. Figure 2 shows the hot deformation process diagram in the present work.

## 3. Results

### 3.1. Stress–Strain Curves

Stress–strain curves are shown in Figure 3. Most of the curves exhibit typical DRX behavior with a single peak stress followed by a gradual fall towards a steady state stress [24]. The main softening mechanism of the dual-scale SiCp/A356 composites is DRX. When the samples deformed at the strain rate of 0.1 s^−1^, the cyclic stress peaks can be observed (Figure 3b). In the process of hot deformation, DRX and work hardening take place at the same time. Under the action of DRX and work hardening, the cyclic stress peaks can be observed as a result of the slow dislocation density change. The peak stress increases when the strain rate increases and the deformation temperature decreases (Table 2). Because of the higher temperature and smaller the strain rate, the deformation resistance of the material is smaller. It is also easier for the composites to achieve steady deformation at high temperature and low strain rate. For example, when the temperature is 520 °C and the strain rate is 0.01 s^−1^, the phenomenon of DRX is not obvious. A plateau in the stress–strain curve is achieved (Figure 3a).

### 3.2. Constitutive Equation

Hot-deformation behavior is a thermal activation process. Constitutive equation is used to calculate the activation energy (*Q*). The power law description of stress (Equation (1)) is preferred for relatively low stresses. Conversely, the exponential law (Equation (2)) is only suitable for high stresses [25,26]. However, the hyperbolic sine law (Equation (3)) can be used for a wide range of temperatures and strain rates [27]. The Zener–Hollomon parameter (*Z*) is the temperature-compensated strain rate (Equation (4)).
(1)$$\dot{\varepsilon }=A_{1}\sigma^{n\prime }\exp(\frac{-Q}{RT})\;\;\;\;\;(\alpha \sigma \;\leq \;0.8)$$
(2)$$\dot{\varepsilon }=A_{2}\exp(\beta \sigma )\exp(\frac{-Q}{RT})\;\;\;\;\;(\alpha \sigma \;\geq \;1.2)$$
(3)$$\dot{\varepsilon }=A{\left[\sinh(\alpha \sigma )\right]}^{n}\exp(\frac{-Q}{RT})\;\;\;\;(\text{for all}\;\alpha \sigma )$$
(4)$$Z=\dot{\varepsilon }\exp(\frac{Q}{RT})=A{\left[\sinh(\alpha \sigma )\right]}^{n}$$
where *A*_1_, *A*_2_, *A*, *α*, *n*, *β* and *n′* are material constant (*α* = *β*/*n′*), *R* is the universal gas constant, *T* is the deformation temperatures (*T* is in K).

By taking a natural logarithm from each side of Equations (1)–(3), Equations (5)–(7) could be derived for the peak stress.
(5)$$\ln\dot{\varepsilon }=\ln A_{2}+n\prime \ln\sigma -\frac{Q}{RT}$$
(6)$$\ln\dot{\varepsilon }=\ln A_{1}+\beta \sigma -\frac{Q}{RT}$$
(7)$$\ln\dot{\varepsilon }=\ln A-\frac{Q}{RT}+n\ln[\sinh(\alpha \sigma )]$$
when the temperature is certain, *n′* and *β* are the slopes of ln$\dot{\varepsilon }$–ln*σ* and ln$\dot{\varepsilon }$–*σ* curves, respectively. By taking partial derivative of Equation (3), Equation (8) is obtained.
(8)$$Q=R\frac{\partial \ln[\sinh(\alpha \sigma )]}{\partial (1/T)}|{}_{\dot{\varepsilon }}\cdot \frac{\partial \ln\dot{\varepsilon }}{\partial \ln\sinh(\alpha \sigma )]}|{}_{{}_{T}}=nRK$$
where *n* and *K* are the slopes of ln$\dot{\varepsilon }$–ln[sinh(*ασ*)] and ln[sinh(*ασ*)]–1/*T* curves, respectively.

It follows from Equations (5) and (6) that the slope of the plot of ln$\dot{\varepsilon }$ against ln*σ* and the slope of the plot of ln$\dot{\varepsilon }$ against *σ* can be used for obtaining the values of *n′* and *β*, respectively (Figure 4a,b). The value of *σ* takes the peak stress. By calculating the average values of the slopes of different straight lines, the average values of 12.78 and 0.1871; for *n′* and *β* can be obtained, respectively. This gives the value of *α = β/n′* = 0.01464. According to Equation (7), the slope of the plot of ln$\dot{\varepsilon }$ against ln[sinh(*ασ*)] and the slope of the plot of ln[sinh(*ασ*)] against 1000/*T* can be used for obtaining the value of *n* and *K*, respectively (Figure 4c,d). The average value of *n* and *K* were determined as 10.2816 and 5.1711, respectively. According to Equation (8), the activation energy can be calculated (*Q = nRK* = 442.03 kJ/mol).

By taking a natural logarithm from each side of Equation (4), Equation (9) could be derived. According to Equations (8) and (9), the value of ln*Z* under the different *T* and $\dot{\varepsilon }$ can be obtained (Table 3). It follows from Equation (9) that the intercept on ln*Z* axis of the plot of ln*Z* against ln[sinh(*ασ*)] can be used for obtaining the value of ln*A*. The value of *A* was determined as *A* = *e*^66.93^. It can be seen in Figure 5 that the plot of ln*Z* against ln[sinh(*ασ*)] are in good agreement with the test data of SiCp/A356 composites under different deformation conditions (Table 3).
(9)$$\ln Z=\ln\dot{\varepsilon }+Q/RT=\ln A+n\ln[\sinh(\alpha \sigma )]$$

Taking the values of *n*, *α*, *A* and *Q* into Equation (3), the constitutive equation of SiCp/A356 composites is obtained as follows:$$\dot{\varepsilon }=e^{66.93}\times {\left[\sinh(0.01464\sigma )\right]}^{10.2816}\exp(-\frac{442.03}{8.314T})$$

The calculated peak stress value by the constitutive equation is compared with the tested value (Figure 6a). The fitting degree is quantitatively evaluated by using the statistical parameters such as linear correlation coefficient (*R*) and average relative error (AARE) (Equations (9) and (10)). The results show that the correlation coefficient is 0.951(*R* > 0.95) and the average relative error is 4.26% (AARE < 5%). This indicates that the constitutive equation has high fitting degree and applicability (Figure 6b). Although it is difficult for the constitutive equation to predict the deformation mechanism of discontinuously reinforced aluminum (DRA) composites, the constitutive equation is the first basis to reveal the complex rheological behavior of DRA composites. The modified equation including the load transfer effect provided a much more detailed description of the creep behavior of the DRA composite [28]. The high temperature creep behavior and hot-deformation behavior of aluminum alloys and its composites can all be regarded as a process of thermal activation energy. The establishment of perfect and accurate constitutive equation is the basis for promoting the development of processing technology of DRA composites. Therefore, the load transfer effect may be used in the hot-deformation behavior.
(10)$$R=\frac{\sum_{i=1}^{N}\left(E_{i}-\bar{E})(P_{i}-\bar{P}\right)}{\sqrt{\sum_{i=1}^{N}{\left(E_{i}-\bar{E}\right)}^{2}\sum_{i=1}^{N}{\left(P_{i}-\bar{P}\right)}^{2}}}=0.951$$
(11)$$AARE=\frac{1}{N}\sum_{i=1}^{N}\left|\frac{E_{i}-P_{i}}{E_{i}}\right|\times 100\%=4.26\%$$
where, *E_i_* is the test stress value, *P_i_* is the calculated stress value, $\bar{E}$ and $\bar{P}$ is the average test value and the calculated value of the stress, respectively. *N* is the number of all data points.

### 3.3. Processing Maps

A processing map is established mainly based on a dynamic material model (DMM) [29], which is composed of a power dissipation diagram and a rheological instability diagram. It is used to describe the safe domains and instability domains. The power dissipation diagram reflects the change of microstructure under certain deformation temperatures and strain rates, such as dynamic recovery, dynamic recrystallization and super plasticity, which is generally expressed by the power dissipation factor *η* [30] (Equation (12)).
(12)$$\eta =\frac{2m}{m+1}$$
where *m* is the strain rate sensitive factor. Under a certain *T* and $\dot{\varepsilon }$, *m* can be expressed as Equation (13):(13)$$m=\frac{\partial \ln\sigma }{\partial \ln\dot{\varepsilon }}$$

The cubic spline function is used to fit the relation curve between ln*σ* and ln$\dot{\varepsilon }$, and then the slope of the spline function is the value of *m*. The value of *η* can be obtained from Equation (12). The contour curve of *η* is drawn in the *T*–ln$\dot{\varepsilon }$ plane. The result shows that the higher *η*, the better the hot performance of the materials. However, the high *η* does not completely indicate good hot performance due to the fact that the value of *η* may be very high in the flow instability region. Therefore, it is necessary to clarify the flow instability region accurately [31].

According to the principle of irreversible thermodynamic extreme value, the criterion of continuous instability plastic deformation of materials can be expressed by dimensionless parameters $\xi (\dot{\varepsilon })$ [32] (Equation (14)):(14)$$\xi (\dot{\varepsilon })=\frac{\partial \ln\left(m/(m+1)\right)}{\partial \ln\dot{\varepsilon }}+m\;<\;0$$

Therefore, the instability condition is also related to *m*. When the power dissipation diagram and rheological instability diagram are superposed, it is the processing map (Figure 7). The region ($\xi (\dot{\varepsilon })$ < 0) is the rheological instability diagram (the gray area in Figure 7). It can be seen that there are three instability domains: the region with strain rate of 0.62–5 s^−1^ and deformation temperature of 464–495 °C; the strain rate of 0.01–0.03 s^−1^ and deformation temperature of 460–512 °C; and the strain rate of 1–5 s^−1^ and deformation temperature of 518–520 °C. In addition, there are two safe processing areas: the area with strain rate of 0.62–5 s^−1^ and deformation temperature of 495–518 °C (DOMI) and the area with strain rate of 0.03–0.62 s^−1^ and deformation temperature of 460–520 °C (DOMII). The DOMI with the highest *η* is the best for the hot processing for dual-scale SiCp/A356 composites. When the deformation temperature is 500 °C, no matter how the strain rate changes in the range of 0.03–5 s^−1^, it is suitable for the hot processing.

### 3.4. Microstructure Analysis

Figure 8 shows the effects of deformation temperature on the microstructure when the strain rate is 1 s^−1^. When the temperatures is 460 °C there are obviously disordered and entangled dislocation lines in the matrix (Figure 8a). When the deformation temperature is 480 °C, DRX grains are generated with clear grain boundaries but part of the grains are in a curved shape (Figure 8b). At the same time, a large number of dislocation structures can be seen inside and outside the recrystallized grains. This indicates that the DRX grains are not fully grown. When the deformation temperature rises above 500 °C (Figure 8c,d), the grain boundary of the DRX grain is straight and clear. The DRX grain is obviously isometric, and the dislocation density in the grain is very low. This indicates that DRX grain has grown sufficiently at 500 °C. Figure 9 shows the effects of strain rate on the microstructure when the deformation temperature is 500 °C. At a high strain rate, the growth of DRX grain is inhibited, and there are some scattered dislocations in the grains (Figure 9a,b). At a low strain rate, dislocation density decreases, grain boundary becomes clear and sharp, and the size of DRX grain increases (Figure 9c,d).

With the increase of deformation temperature and the decrease of the strain rate, dislocation density around the SiC particles gradually reduces due to the easy climbing and slip of the dislocation. Moreover, the DRX grain grew up gradually, and the DRX grain boundary gradually clears and straightens [33,34]. DRX grain size increases significantly when the deformation temperature is up to 520 °C from 500 °C. This shows that the effect of deformation temperature on the dynamic recrystallization of materials is more obvious than that of strain rate. This is also consistent with the optimal processing conditions of materials, where the temperature range is narrow (495–518 °C) and the strain rate range is wide (0.62–5 s^−1^) (Figure 7).

Moreover, DRX grains mainly occur around SiC particles. Therefore, the presence of SiC particles can promote effectively the DRX nucleation. The number of DRX grains in the “safe” domains (Figure 8c and Figure 9c) is larger and the dislocation density is lower compared to those of instability domains. The typical microstructural characteristics of the “safe” domains correspond to the SiC particles and hot-processing parameters.

### 3.5. Hot-Deformation Behavior

In general, the hot-deformation activation energy *Q* is an important physical parameter that qualitatively reflects the energy barrier for dislocation motion during hot deformation. Cavaliere P. mentions the flow stress behavior of DRA composite is government by two main processes: The transfer of load from the ductile matrix to the hard particles and the microstructural transformations such as DRX or damage phenomena; in this case the materials can present decohesion at the matrix–particles interfaces or several particles cracking [28,35]. Hot deformation behavior of the dual-scale SiCp/A356 composites can be explained by model of dislocation particulate interactions (Figure 10). In the process of the hot deformation, the dislocation in the aluminum matrix slips under the action of the external stress, while SiC particles as a non-deformable phase is an obstacle to dislocation motion. When the dislocation moves to the particle, stress concentration is generated. The stress also can be relaxed due to the climbing of the dislocation. Whether or not the dislocation will accumulate at the particle and how much it accumulates depends on the rate of the dislocation arriving at the particle (*R1*) and the climbing rate of the dislocation (*R2*). *D* is used to represent the dislocation accumulation at the particle, and the following functional relationship exists (Equation (15)):*D* = *f*(*R1*,*R2*)(15)
where *R1* is related to the strain rate during the hot deformation, *R2* is related to the deformation temperature, particle size and deformation mechanism. When the value of *R2* is greater than the value of *R1*, the dislocation will not accumulate around the SiC particle (Figure 10c). The material will have a lower work-hardening rate and flow stress. When the value of *R2* is smaller than the value of *R1*, the dislocation will accumulate around the SiC particle (Figure 10d). A complex dislocation structure will be formed at the particle with the increase of the accumulation degree. The material shows high machining hardening rate and flow stress [22].

At the early stage of deformation, *R1* can be considered to be constant for the given initial rate (*R1* > *R2*). At this time, with the increase of deformation degree, the dislocation accumulation at the SiC particles leads to the increase of the hardening rate and stress, which is more significant at low temperature deformation (Figure 3a). With the further increase of deformation degree, DRX occurs. Dislocation at the SiC particle plugging area gradually tends to maintain a stable state. At this time, the flow stress tends to be stable. It is consistent with the stress–strain curve in Figure 3. With the increase of the deformation temperature (the value of *R2* increases), the dislocation density around the SiC particle decreased, so the peak stress decreased. However, with the increase of the strain rate (as the value of *R1* increases), the dislocation density at the particle increased, so the peak stress of the material increased. This is consistent with the peak stress in Table 2. Although, it is proposed that aluminum and its composites are difficult to produce by DRX. Since, the climb and cross-slip of the dislocation are easy due to the high stacking-fault energy of this kind of material. The main softening mechanism of the dual-scale SiCp/A356 composites is DRX. It is indicated that the dual-scale SiCp/A356 composites can obtain a homogenous ultrafine grained structure via hot deformation. The average *Q* value for the dual-scale SiCp/A356 composites is found to be 442.03 kJ/mol. The average activation energy for the nano-SiCp/Al–Si composites was 277 kJ/mol, which was larger than the activation energy for self-diffusion of pure aluminum (144 kJ/mol). The high *Q* value for nano-sized SiCp/Al–Si composites can be attributed to the nano-sized SiC hindering dislocations and the growth of grain boundaries [16]. It is worth mentioning that the *Q* value of the dual-scale SiCp/A356 composites is much higher than the value for 15 vol.% SiC/A356 (263 kJ/mol) and A356 (161 kJ/mol) [36]. Chen X.R et al. found that the *Q* value of Al-7 wt.% Si alloy reinforced by 10 wt.% micro-sized SiCp and 5 wt.% nano-sized TiB2 is 269.7 kJ/mol, which is much higher than the value for Al–Si alloy. The higher value of activation energy can be attributed to the addition of TiB2 reinforcements as a result of their serious retardation of dislocation movement in hot deformation. It should be mentioned that the size of hard particles, the inter-particle spacing and their morphologies can affect the conditions for the occurrence of DRX. Irregular SiC particles with a large size are prone to breakage during deformation, which leads to a rotation of the broken particles and a weak interface between the particles and the Al matrix, and relaxes the stress concentration, while the presence of TiB_2_ nanoparticles leads to typical strain hardening due to their pinning effect on dislocations [18]. Moreover, the *Q* value for nano-SiCp/Al–Si composites was slightly larger than that for Al–Si composites reinforced by micro-SiC, which may be attributed to the nano-SiC retarding the occurrence of the softening mechanism by hindering the growth of grain boundaries or subgrains [16].

## 4. Conclusions

Hot-deformation behavior and the corresponding microstructures of the dual-scale SiCp/A356 composites were investigated in the temperature range of 460–520 °C and the strain rate range of 0.01–5 s^−1^. The following conclusions have been drawn:Sress–strain curves showed typical DRX with a single peak stress followed by a gradual fall towards steady-state stress. Peak stress decreased with the increase of deformation temperature and the decrease of strain rate.The average deformation activation energy was 442.03 kJ/mol. The correlation coefficient R for linear regression of constitutive model was 0.951, which indicated the high accuracy of this model. The constitutive equation was expressed as: $\dot{\varepsilon }=e^{66.93}\times {\left[\sinh(0.01464\sigma )\right]}^{10.2816}\exp(-\frac{442.03}{8.314T})$There are two “safe” domains based on the processing maps: one is at temperatures of 495–518 °C and strain rate of 0.62–5 s^−1^; another is at temperatures of 460–520 °C and strain rate of 0.03–0.62 s^−1^. When the temperature is 495–518 °C and strain rate is 0.62–5 s^−1^, it is suitable for the hot processing of dual-scale SiCp/A356 composites.The deformed microstructures at the corresponding processing parameters mainly exhibit DRX. The number of DRX grains in the “safe” domains is larger and the dislocation density is lower compared to those of instability domains. The typical DRX microstructural characteristics of the “safe” domains correspond to the SiC particles and hot-processing parameters.DRX grains mainly occur around SiC particles. The presence of SiC particles can promote effectively the DRX nucleation, which results in the dynamic softening mechanism of the dual-scale SiCp/A356 composites being dominated by DRX.

## Figures and Tables

**Figure 1 materials-13-02825-f001:**
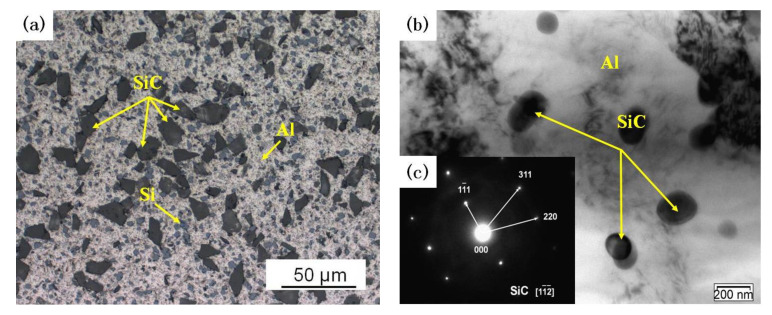
(**a**) Optical microstructure and (**b**) transmission electron microscopy (TEM) images of the composites; (**c**) diffraction pattern of the nano SiC.

**Figure 2 materials-13-02825-f002:**
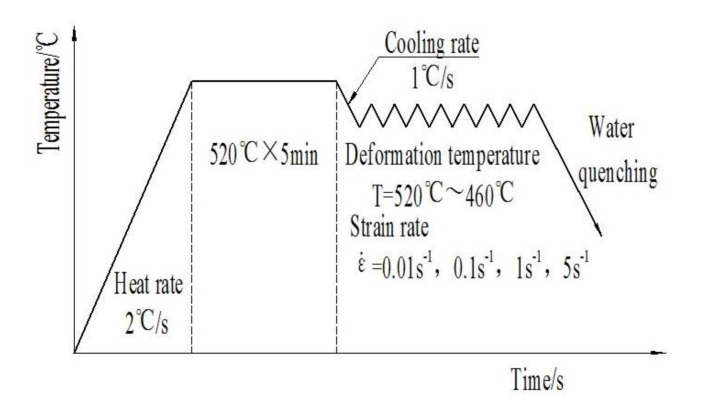
The hot-deformation process diagram.

**Figure 3 materials-13-02825-f003:**
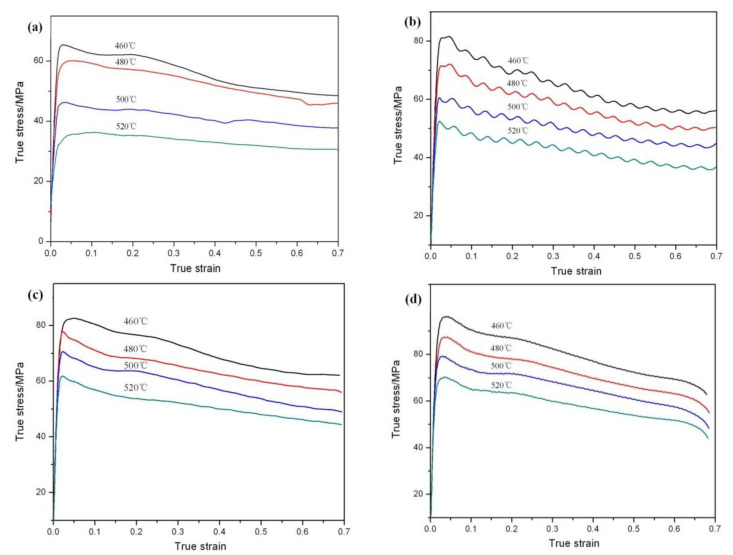
True strain–stress curves at different strain rates: (**a**) 0.01 s^−1^; (**b**) 0.1 s^−1^; (**c**) 1 s^−1^; (**d**) 5 s^−1^.

**Figure 4 materials-13-02825-f004:**
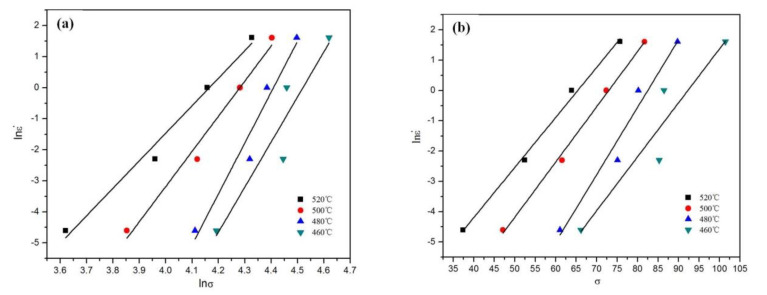
Plots used for calculation of hot-processing parameters. (**a**) ln$\dot{\varepsilon }$–ln*σ*, (**b**) ln$\dot{\varepsilon }$–*σ*, (**c**) ln$\dot{\varepsilon }$–ln[sinh(*ασ*)], (**d**) ln[sinh(*ασ*)]–1000/*T*.

**Figure 5 materials-13-02825-f005:**
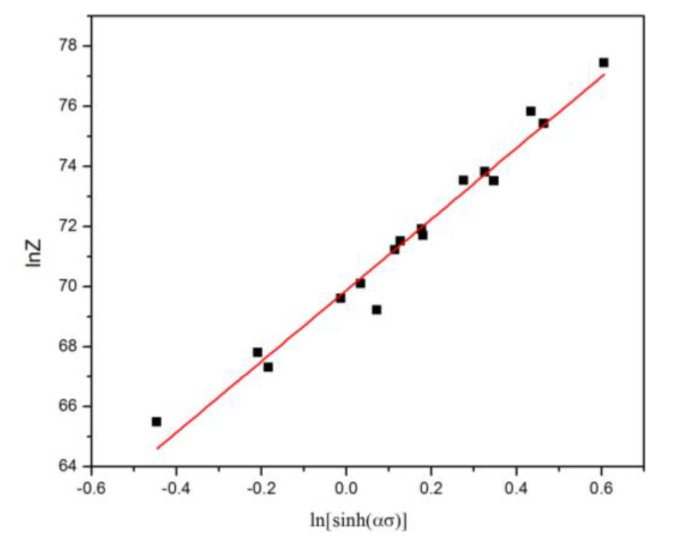
Plots used to derive the constitutive equations.

**Figure 6 materials-13-02825-f006:**
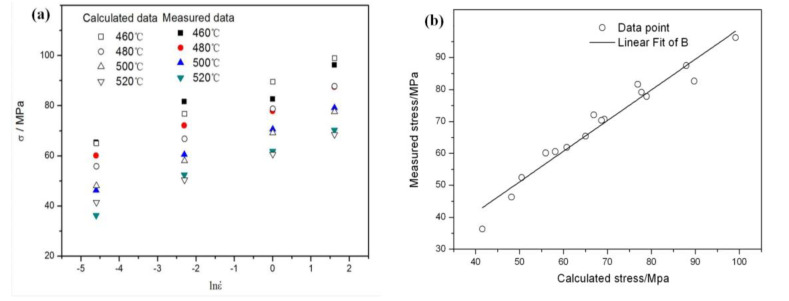
(**a**) The comparison and (**b**) Correlation between the calculated and measured values of peak stress for the dual-scale SiCp/A356 composites.

**Figure 7 materials-13-02825-f007:**
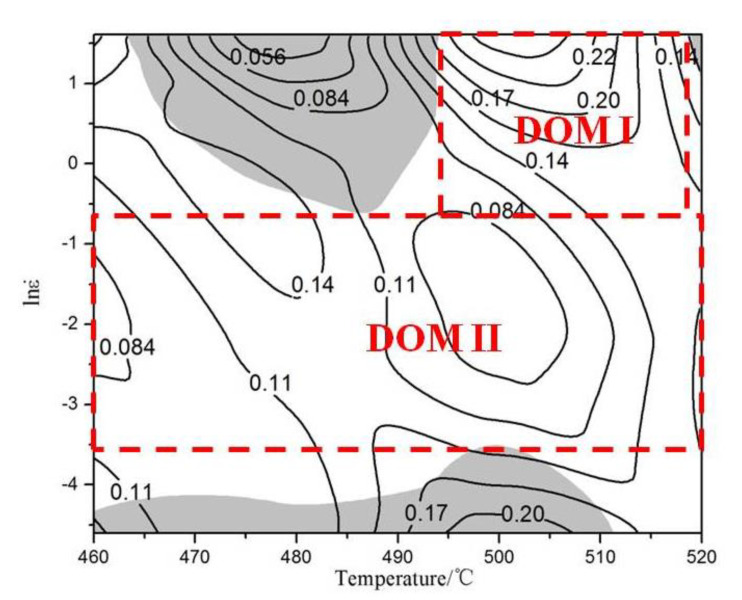
Processing maps of the dual-scale SiCp/A356 composites at the strain of 0.5.

**Figure 8 materials-13-02825-f008:**
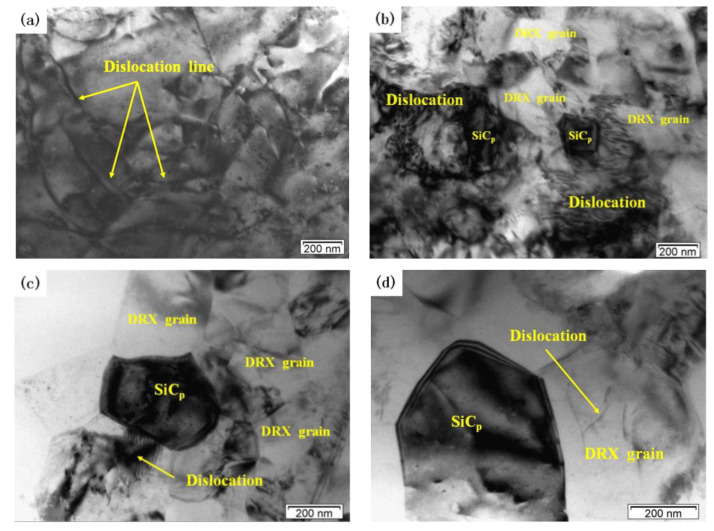
TEM images of the dual-scale SiCp/A356 composites deformed at the strain rate of 1 s^−1^ and different deformation temperatures (**a**) 460 °C; (**b**) 480 °C; (**c**) 500 °C; (**d**) 520 °C.

**Figure 9 materials-13-02825-f009:**
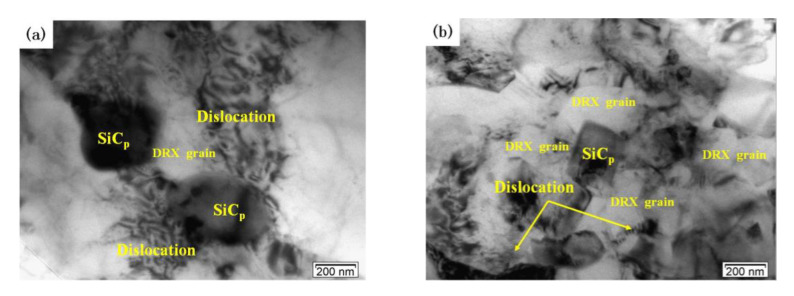
TEM images of the dual-scale SiCp/A356 composites deformed at 500 °C and different strain rates: (**a**) 5 s^−1^; (**b**) 1 s^−1^; (**c**) 0.1 s^−1^; (**d**) 0.01 s^−1^.

**Figure 10 materials-13-02825-f010:**
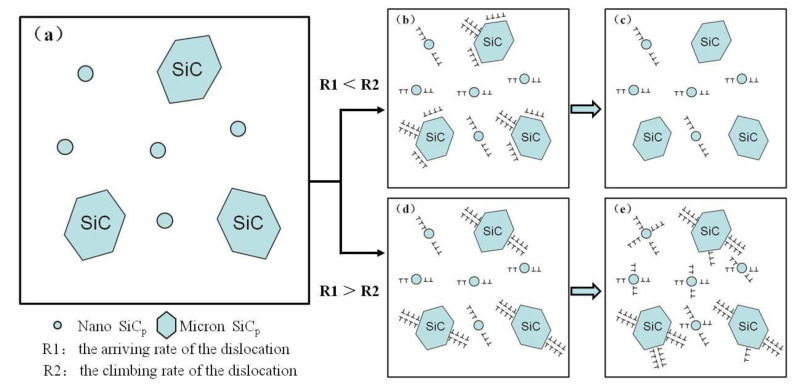
Model of dislocation particulate interactions of the dual-scale SiCp/A356 composites. (**a**) the initial microstructure, (**b**) dislocations pile-up around SiC particles, (**c**) dislocation climb, (**d**) dislocation accumulate, (**e**) dislocation structure with the increase of the dislocation accumulation degree.

**Table 1 materials-13-02825-t001:** Chemical compositions of the A356 alloy powder (wt. %).

Element	Si	Mg	Cu	Fe	Al
Weight Percent	7.0	0.3	0.1	0.1	Balance

**Table 2 materials-13-02825-t002:** Peak stress at different hot-processing parameters (MPa).

Strain Rate/s^−1^	Temperature/°C			
	460	480	500	520
0.01	65.37618	60.13297	46.28657	36.30753
0.1	81.61172	72.08328	60.55818	52.44261
1	82.60639	77.8271	70.65215	61.87151
5	96.25351	87.496	79.18527	70.3212

**Table 3 materials-13-02825-t003:** The value of ln*Z*.

$\dot{\varepsilon }/s^{-1}$	*T*/°C			
	460	480	500	520
0.01	67.92850207	66.00197691	64.17514257	62.44045625
0.1	70.23108716	68.304562	66.47772766	64.74304134
1	72.53367225	70.60714709	68.78031276	67.04562643
5	74.14311016	72.216585	70.38975067	68.65506434

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
