# Peer review of "Hot-Deformation Behavior and Microstructure Evolution of the Dual-Scale SiCp/A356 Composites Based on Optimal Hot-Processing Parameters"

_materials, 2020, doi:10.3390/ma13122825_

Round 1
Reviewer 1 Report
The present work is an interesting study on hot deformation of SiCp/A356 metal matrix composite. The composite is fabricated via powder metallurgical route adding both micro and nano SiC particles to the aluminum matrix. Both kind of particles greatly influence composite mechanical behavior. The mechanical behavior of the manufactured composite is studied at high temperature and different strain rates. The strain rate is modeled using the Zener-Hollomon parameter. An activation energy for hot deformation of 442kJ/mol is founds. This value is really high in comparison to aluminum self-diffusion energy. The mechanical behavior is described by a combination of accumulation and recovery of dislocations. DRX is the main mechanism explaining the mechanical behavior.
Finally, a processing map showing the ideal conditions for hot deformation of the studied composites is provided showing a range of optimal processing parameters.
The work presents two aspects that should be improved and well justified:
1.- The load transfer effect must be considered in the description of the high temperature behavior of the studied composite. The load transfer effect during loading of metal matrix composites, even at high temperature, is well known. In particular, the effect of the load transfer effect both on n (stress exponent) and Q (activation energy) is described in detail in “Threshold stress and load partitioning during creep of metal matrix composites. R. Fernández and G. González-Doncel. Acta Mater 56 (2008) 2549-2562”. A modified equation (page 6 line 171) including the load transfer effect would provide a much more detailed description of the deformation behavior of the studied composite at high temperature.
2.- The relevance of the present work mainly relies on the addition of micro and nano sized SiC particles. This is very interesting. However, there is no description in the work about the size effect of SiC particles. In the sketch shown in figure 10 both kind of particles are considered equivalent in relation to the interaction with dislocations. However, the difference in size between micro and nano particles (around one order of magnitude) must be described in detail.
Some important references in the field of mechanical behavior of metal matrix composites are missing.
Author Response
Manuscript number: Materials-821092
MS Type: Article
Title: Hot deformation behavior and microstructure evolution of the dual-scale SiCp/A356 composites based on optimal hot processing parameters
Correspondence Author: Aiqin Wang
Dear Editor and Reviewer,
We quite appreciate your favorite consideration and the reviewer’s insightful comments. We tried our best to improve the manuscript, and have revised the manuscript according to your kind advice and reviewer’s detailed suggestions. And found these comments are all valuable and very helpful for revising and improving our paper, as well as the important guiding significance to our research. Revised portion are made in red in paper. Enclosed please find the responses to the reviewers. We sincerely hope this manuscript will be finally acceptable to be published on Materials. Thank you very much for all your help and looking forward to hearing from you soon.
With best regards,
Yours sincerely
Corresponding Author: Aiqin Wang
Response to the reviewer’s comments
Reviewer #1:
The present work is an interesting study on hot deformation of SiCp/A356 metal matrix composite. The composite is fabricated via powder metallurgical route adding both micro and nano SiC particles to the aluminum matrix. Both kind of particles greatly influence composite mechanical behavior. The mechanical behavior of the manufactured composite is studied at high temperature and different strain rates. The strain rate is modeled using the Zener-Hollomon parameter. An activation energy for hot deformation of 442kJ/mol is founds. This value is really high in comparison to aluminum self-diffusion energy. The mechanical behavior is described by a combination of accumulation and recovery of dislocations. DRX is the main mechanism explaining the mechanical behavior.
Finally, a processing map showing the ideal conditions for hot deformation of the studied composites is provided showing a range of optimal processing parameters.
The work presents two aspects that should be improved and well justified:
Comment 1.The load transfer effect must be considered in the description of the high temperature behavior of the studied composite. The load transfer effect during loading of metal matrix composites, even at high temperature, is well known. In particular, the effect of the load transfer effect both on n (stress exponent) and Q (activation energy) is described in detail in “Threshold stress and load partitioning during creep of metal matrix composites. R. Fernández and G. González-Doncel. Acta Mater 56 (2008) 2549-2562”. A modified equation (page 6 line 171) including the load transfer effect would provide a much more detailed description of the deformation behavior of the studied composite at high temperature.
Response: Thanks for the Reviewer’s kind suggestion. In our research, we mainly focus on the hot working behaviors of the composite materials under different processing conditions in order to optimize the processing parameters and control their microstructures. And the hot compression tests were taken by Gleeble-1500D thermal simulator. The true stress-strain curves can be obtained. According to true stress-strain curves, the hot working behaviors of the composite materials can be described. As mentioned in the article (Threshold stress and load partitioning during creep of metal matrix composites), load transfer effect must be considered in the description of the high temperature behavior of the studied composite. Thank you very much for providing us with research directions. And we are also very interested in the creep behavior. The creep behavior is essential to characterize the performance of the material at high temperature. It would be the new direction for future researches. Thanks for your share.
Comment 2.The relevance of the present work mainly relies on the addition of micro and nano sized SiC particles. This is very interesting. However, there is no description in the work about the size effect of SiC particles. In the sketch shown in figure 10 both kind of particles are considered equivalent in relation to the interaction with dislocations. However, the difference in size between micro and nano particles (around one order of magnitude) must be described in detail.
Some important references in the field of mechanical behavior of metal matrix composites are missing.
Response: Thanks for the Reviewer’s kind suggestion. DRX is related to true stress, which is related to dislocation density. While the dislocation density is related to distortion energy, which is the driving force of recrystallization. At the same hot deformation condition and prepared method of the composites, the peak stress can reveals the DRX. The peak stress of 2 vol % nano-SiCp/A356 composite is in the range of 25-70MPa. And the peak stress of the dual-scale SiCp/A356 composites (content of micro- and nano-sized SiC particles is 2 vol.% and 13 vol.%, respectively.), is in the range of 36-96 MPa. But these differences were not very significant. We can conclude that the effects of 2 vol. % nano-sized SiCp and 13 vol. % micro-sized SiC particles on the DRX is in the same order of magnitude.
In this research, we mainly focus on the hot working behaviors of the composite. The mechanical behavior of metal matrix composites will be analyzed in the future researches.
In all, we found the reviewer’s comments are quite helpful and valuable in improving the quality of our manuscript, and we revised our paper point-by-point. Thank you and the reviewer again for your help!
Reviewer 2 Report
General comment on English: the manuscript must be deeply revised in order to improve English. Just for example “Line 49 – There are many research work”, “Line 54 – Its effect… are”, “Line 98 - Fig. 2 show” and so on…
The acronim DRX is not defined at the beginning but only at line 53.
Line 34-39 the sentence “however … remarkable” is too long and obscure.
Line 49 has no sense “there are many research work have been conducted to study”
Line 60-62 “The non-uniform deformation areas around the large and widely spaced micron 60 SiC particles with high density dislocation and the large lattice orientation difference with the matrix 61 leading to the particle induced nucleation (PSN)” has no sense.
Table 1: Al to balance, not others.
Line 112 “This is due to the 112 recrystallization soften effect cannot be balanced with the new work hardening because of the low 113 dislocation increment rate.” What does it mean???
Line 116 Strain not train.
Line 118 “the DRX is not obvious”. What does it mean???
By the way the manuscript is very similar to an earlier one already published on Materials 2020 13(8), 1812, cited in [16]. Also the material (A356 – AlSi alloy), reinforcement (SiC), and experimental technique are quite similar (except for TEM analysis).
In literature there are many papers on the same topic and it is not clear in particular what is really new:
Constitutive Equation and Processing Maps of Al-7Si-0.3 Mg Hybrid Composites: a Novel Approach to Reduce Cost of Material by Using Agro-Industrial Wastes. Silicon 11 2633-2646 (2019) https://doi.org/10.1007/s12633-018-0053-4;
Hot workability of aluminum particulate composites by Paola Leo, Emanuela Cerri, Hugh McQueen;
Proceedings: Metal matrix composites and physical properties, Volume 3 1997 pag. 420
The manuscript may be reconsidered after a deep revision especially on the motivation and the overall scientific soundness respect to the state of the art.
Author Response
Manuscript number: Materials-821092
MS Type: Article
Title: Hot deformation behavior and microstructure evolution of the dual-scale SiCp/A356 composites based on optimal hot processing parameters
Correspondence Author: Aiqin Wang
Dear Editor and Reviewer,
We quite appreciate your favorite consideration and the reviewer’s insightful comments. We tried our best to improve the manuscript, and have revised the manuscript according to your kind advice and reviewer’s detailed suggestions. And found these comments are all valuable and very helpful for revising and improving our paper, as well as the important guiding significance to our research. Revised portion are made in red in paper. Enclosed please find the responses to the reviewers. We sincerely hope this manuscript will be finally acceptable to be published on Materials. Thank you very much for all your help and looking forward to hearing from you soon.
With best regards,
Yours sincerely
Corresponding Author: Aiqin Wang
Response to the reviewer’s comments
Reviewer #2:
Comment 1. General comment on English: the manuscript must be deeply revised in order to improve English. Just for example “Line 49 – There are many research work”, “Line 54 – Its effect… are”, “Line 98 - Fig. 2 show” and so on…
Response: Thanks for the Reviewer’s kind suggestion. We are sorry for the syntax error. The manuscript was deeply revised in order to improve English. And the revised portion was made in red in paper.
Comment 2. The acronim DRX is not defined at the beginning but only at line 53.
Response: Thanks for the Reviewer’s kind suggestion. We are really sorry for this kind of mistake. The acronim DRX is defined at line 22.
Comment 3. Line 34-39 the sentence “however … remarkable” is too long and obscure.
Response: Thanks for the Reviewer’s kind suggestion. Line 34-39 the sentence “however … remarkable” is changed to “And the properties of SiCp/Al-Si composites are depended on the amount and size of SiC particles. When the SiC particles amount is determined, the micro-sized SiC particles could improve the hardness and strength of Al-Si matrix composites at the expense of plasticity. While the nano-sized SiC particles could improve the plasticity of Al-Si matrix composites, the enhancement of strength and wear resistance is not remarkable.”
Comment 4. Line 49 has no sense “there are many research work have been conducted to study”.
Response: Thanks for the Reviewer’s kind suggestion. Line 49 “there are many research work have been conducted to study” is changed to “researches are mainly focused on the hot deformation behavior of either micro-sized or nano-sized SiCp/Al composite”.
Comment 5. Line 60-62 “The non-uniform deformation areas around the large and widely spaced micron 60 SiC particles with high density dislocation and the large lattice orientation difference with the matrix 61 leading to the particle induced nucleation (PSN)” has no sense.
Response: Thanks for the Reviewer’s kind suggestion. Line 60-62 is changed to “High density dislocations and large lattice orientation difference with the matrix occurred in the non-uniform deformation areas around the large and widely spaced micro-sized SiC particles, which is leading to the particle induced nucleation (PSN)”
Comment 6. Table 1: Al to balance, not others.
Response: Thanks for the Reviewer’s kind suggestion. It has been modified.
Comment 7. Line 112 “This is due to the 112 recrystallization soften effect cannot be balanced with the new work hardening because of the low 113 dislocation increment rate.” What does it mean???
Response: Thanks for the Reviewer’s question. DRX is the main softening mechanism. In the process of hot deformation, DRX and work hardening took place at the same time. Under the action of DRX and work hardening, the cyclic stress peaks can be observed as a result of the slow dislocation density change (strain rate is 0.1 s-1). This is a typical deformation behavior of metals. And it has been corrected to make them easy to understand.
Comment 8. Line 116 Strain not train.
Response: Thanks for the Reviewer’s kind suggestion. We are really sorry for this kind of mistake.
Comment 9. Line 118 “the DRX is not obvious”. What does it mean???
Response: Thanks for the Reviewer’s question. It means “the phenomenon of DRX is not obvious”. The reduced level of stress can reflect the degree of DRX. When the temperature is 520℃ and the strain rate is 0.01 s-1, a plateau in the stress-strain curve is achieved(Figure 3a). It indicated that the phenomenon of DRX is not obvious.
Comment 10. By the way the manuscript is very similar to an earlier one already published on Materials 2020 13(8), 1812, cited in [16]. Also the material (A356 – AlSi alloy), reinforcement (SiC), and experimental technique are quite similar (except for TEM analysis).
Response: Thanks for the Reviewer’s question. We are in the same research group with the writer of the manuscript published on Materials 2020 13(8), 1812, cited in [16]. The content and size of SiC particles are the great differences between these two papers. The manuscript cited in [16] researches the hot deformation behavior of 2 vol % nano-SiCp/Al-Si composite. While in this paper the hot deformation behavior and microstructure evolution of the dual-scale SiCp/A356 composites was researched. The content of micro- and nano-sized SiC particles in the composite was 2 vol.% and 13 vol.%, respectively. The true strain–stress curves of 2 vol % nano-SiCp/Al-Si composite were shown below. It indicated that the deformation behavior of SiCp/Al composites is greatly influenced by the SiC particle condition (particle size and content) compared with strain–stress curves in this paper. The material (A356 – AlSi alloy), reinforcement (SiC), and experimental technique are similar, but the topics are inherently different.
The true strain–stress curves of 2 vol % nano-SiCp/Al-Si composite
Figure 3. True strain–stress curves at different strain rates: (a) 0.01 s-1; (b) 0.1 s-1; (c) 1 s-1; (d) 5 s-1
The true strain–stress curves of the dual-scale SiCp/A356 composites
Comment 11. In literature there are many papers on the same topic and it is not clear in particular what is really new:
Constitutive Equation and Processing Maps of Al-7Si-0.3 Mg Hybrid Composites: a Novel Approach to Reduce Cost of Material by Using Agro-Industrial Wastes. Silicon 11 2633-2646 (2019) https://doi.org/10.1007/s12633-018-0053-4;
Hot workability of aluminum particulate composites by Paola Leo, Emanuela Cerri, Hugh McQueen;
Proceedings: Metal matrix composites and physical properties, Volume 3 1997 pag. 420
The manuscript may be reconsidered after a deep revision especially on the motivation and the overall scientific soundness respect to the state of the art.
Response: Considering the Reviewer’s suggestion, we have read these two articles carefully. The preparation method is a great difference between these three papers. The aluminium with waste particles (agro and industrial) is prepared by using double stir-casting process (Constitutive Equation and Processing Maps of Al-7Si-0.3 Mg Hybrid Composites: a Novel Approach to Reduce Cost of Material by Using Agro-Industrial Wastes). And Metal-matrix composites (MMC) of 6061, 7075, 2618 and A356 alloys with Al2O3 or SiC particles (15-30 µm) were produced by liquid metal mixing double stir-casting process (Hot workability of aluminum particulate composites). In our research the dual-scale SiCp/A356 composite was prepared via powder metallurgy(PM) by mixing micro-sized SiC powder with the average size of 10 μm, nano-sized SiC powder with the average size of 80 nm, and A356 alloy powder with the average size of 7 μm. The preparation method of particle reinforced aluminum matrix composites has a great influence on the properties of the materials. And the study and develop on the effective preparation process always one of the most important question in research of particle reinforced aluminum matrix composites. The matrix material (A356 – AlSi alloy) is similar, but the topics are inherently different.
In all, we found the reviewer’s comments are quite helpful and valuable in improving the quality of our manuscript, and we revised our paper point-by-point. Thank you and the reviewer again for your help!
Reviewer 3 Report
The authors report on the investigation of the hot deformation behaviour and microstructural characterization of SiCp/A356 composite under different conditions of temperature and strain rate, finding constitutive equations and safe domains for the hot processing of the dual-scale composite.
The reviewer suggests to accept the article after minor revisions, as reported in the following:
Line 77: the authors report on the adoption of specific percentages of SiC particles to be added for the preparation of the composite (i.e. 2% and 13% for micro- and nano-sized particles). Please, clarify the choice of such values.
Lines 96-99: how many replications have been carried out for the hot compression tests? only one replica? Please, explain.
Line 123: please, provide the unit of measure for the peak stress in Table 2.
Lines 141-142: please, provide a better formatting for Equations 6 and 7 (some parts are overlapped).
Line 266: "n" is reported, but in Equation 15 there is "N".
Some minor spell check:
Line 49: please, remove "there are".
Line 117: "high" instead of "highe".
Author Response
Manuscript number: Materials-821092
MS Type: Article
Title: Hot deformation behavior and microstructure evolution of the dual-scale SiCp/A356 composites based on optimal hot processing parameters
Correspondence Author: Aiqin Wang
Dear Editor and Reviewer,
We quite appreciate your favorite consideration and the reviewer’s insightful comments. We tried our best to improve the manuscript, and have revised the manuscript according to your kind advice and reviewer’s detailed suggestions. And found these comments are all valuable and very helpful for revising and improving our paper, as well as the important guiding significance to our research. Revised portion are made in red in paper. Enclosed please find the responses to the reviewers. We sincerely hope this manuscript will be finally acceptable to be published on Materials. Thank you very much for all your help and looking forward to hearing from you soon.
With best regards,
Yours sincerely
Corresponding Author: Aiqin Wang
Response to the reviewer’s comments
Reviewer #3:
The authors report on the investigation of the hot deformation behavior and microstructural characterization of SiCp/A356 composite under different conditions of temperature and strain rate, finding constitutive equations and safe domains for the hot processing of the dual-scale composite.
The reviewer suggests to accept the article after minor revisions, as reported in the following:
Comment 1. Line 77: the authors report on the adoption of specific percentages of SiC particles to be added for the preparation of the composite (i.e. 2% and 13% for micro- and nano-sized particles). Please, clarify the choice of such values.
Response: Thanks for the Reviewer’s question. The content of nano-sized particles was based on previous experimental research. This is one of the experiments being carried on in our lab. According to the previous experimental research, when the volume fraction of nano-sized SiC particles is 1 vol%, 2 vol %, 3 vol % and 4 vol %, the tensile strength of composites improved by 7%, 26%, 14% and 9% respectively, and the yield strength improved by 13%, 43%, 19% and 16% respectively. In our general experimental system, the total volume fraction of SiC particle was designed as 0 vol %,15 vol %, 20 vol %, 25 vol % and 30 vol % (nano-sized SiC particles is 2 vol %) to analysis the effect of SiC particle content on the performance and hot deformation behavior of the composite. And the composite with low volume fraction of SiC (SiC particles content below 40 vol %) was mainly used in automotive wear resistant parts.
Comment 2. Lines 96-99: how many replications have been carried out for the hot compression tests? only one replica? Please, explain.
Response: Thanks for the Reviewer’s kind suggestion. At least 2 replications have been carried out for the hot compression tests. And we found the change rule of the stress-strain curve is the same. Although the value of stress with slightly different, it has little impact on subsequent calculations.
Comment 3. Line 123: please, provide the unit of measure for the peak stress in Table 2.
Response: Thanks for the Reviewer’s kind suggestion. The unit for the peak stress has added.
Comment 4. Lines 141-142: please, provide a better formatting for Equations 6 and 7 (some parts are overlapped).
Response: Thanks for the Reviewer’s kind suggestion. We are really sorry for this kind of mistake. Equations 6 and 7 have been adjusted.
Comment 5. Line 266: "n" is reported, but in Equation 15 there is "N".
Response: Thanks for the Reviewer’s kind suggestion. We are really sorry for this kind of mistake. N(Line 182) is the number of all data points. While in Line 266 and equation (15), it means the dislocation accumulation at the particle. The dislocation accumulation at the particle expressed in D. So, "n" should be changed to "D"(Line 266). The equation (15) has been modified to D.
Comment 6. Some minor spell check:
Line 49: please, remove "there are".
Line 117: "high" instead of "highe".
Response: Thanks for the Reviewer’s kind suggestion. We are really sorry for this kind of mistake. And they all are corrected.
In all, we found the reviewer’s comments are quite helpful and valuable in improving the quality of our manuscript, and we revised our paper point-by-point. Thank you and the reviewer again for your help!
Reviewer 4 Report
The authors of the manuscript have performed a nice research of hot processing of dual-scale A356-SiC composites. Overall, the paper is clearly written and presents novel and interesting results with appropriate graphic representation of the findings. However, I have several minor recommendations, which if reflected in the manuscript may improve its quality. They go as follows:
- In Introduction, paragraph about dynamic recrystallization is not clearly written, especially from P2L60. I advise rewriting of this paragraphs to make connection between various processes happening in the composite more understandable. It may be useful for you to explain in this paragraph why it is necessary to suppress DRX.
- Can you explain your choice of contents of micro- and nanosized SiC particles in the composites? Was it based on previous experimental research or on some theoretical assumptions?
- Check formatting of equations (6) and (7). In my version of the manuscript defects of formatting made it difficult to read the equations, especially equation (7).
- You have determined parameter A to be 66.93. Putting it into equation on P6L171 as you did yield enormously high values for strain rate, about 1030. Please make sure that there is no mistake in your calculations or use of the formulae.
- Figure 7 and P8L206. You have claimed on P8L206 that region with ξ<0 is rheological instability region; however, in figure 7 regions marked as instability region have obviously values of ξ>0 as follows from the isolines. Is there no mistake.
- Figure 7 again. Text below the figure says that the processing map is for fixed strain rate 0.5 s-1, but the figure itself shows a range of strain rates (vertical axis).
- Variable N in text is lowercase, while in equation (15) it is uppercase. Is it the same variable? If it is, please make sure to use the same case in both instances.
Author Response
Manuscript number: Materials-821092
MS Type: Article
Title: Hot deformation behavior and microstructure evolution of the dual-scale SiCp/A356 composites based on optimal hot processing parameters
Correspondence Author: Aiqin Wang
Dear Editor and Reviewer,
We quite appreciate your favorite consideration and the reviewer’s insightful comments. We tried our best to improve the manuscript, and have revised the manuscript according to your kind advice and reviewer’s detailed suggestions. And found these comments are all valuable and very helpful for revising and improving our paper, as well as the important guiding significance to our research. Revised portion are made in red in paper. Enclosed please find the responses to the reviewers. We sincerely hope this manuscript will be finally acceptable to be published on Materials. Thank you very much for all your help and looking forward to hearing from you soon.
With best regards,
Yours sincerely
Corresponding Author: Aiqin Wang
Response to the reviewer’s comments
Reviewer #4:
The authors of the manuscript have performed a nice research of hot processing of dual-scale A356-SiC composites. Overall, the paper is clearly written and presents novel and interesting results with appropriate graphic representation of the findings. However, I have several minor recommendations, which if reflected in the manuscript may improve its quality. They go as follows:
Comment 1. In Introduction, paragraph about dynamic recrystallization is not clearly written, especially from P2L60. I advise rewriting of this paragraphs to make connection between various processes happening in the composite more understandable. It may be useful for you to explain in this paragraph why it is necessary to suppress DRX.
Response: Thanks for the Reviewer’s kind suggestion. The paragraph was deeply revised. And the revised portion was made in red in paper.
Comment 2. Can you explain your choice of contents of micro- and nanosized SiC particles in the composites? Was it based on previous experimental research or on some theoretical assumptions?
Response: Thanks for the Reviewer’s kind suggestion. The content of nano-sized particles was based on previous experimental research. This is one of the experiments being carried on in our lab.According to the previous experimental research, when the volume fraction of nano-sized SiC particles is 1 vol%, 2 vol %, 3 vol % and 4 vol %, the tensile strength of composites improved by 7%, 26%, 14% and 9% respectively, and the yield strength improved by 13%, 43%, 19% and 16% respectively. In our general experimental system, the total volume fraction of SiC particle was designed as 0 vol %,15 vol %, 20 vol %, 25 vol % and 30 vol % (nano-sized SiC particles is 2 vol %) to analysis the effect of SiC particle content on the performance and hot deformation behavior of the composite. And the composite with low volume fraction of SiC (SiC particles content below 40 vol %) was mainly used in automotive wear resistant parts.
Comment 3. Check formatting of equations (6) and (7). In my version of the manuscript defects of formatting made it difficult to read the equations, especially equation (7).
Response: Thanks for the Reviewer’s kind suggestion. We are really sorry for this kind of mistake. Equations 6 and 7 have been adjusted.
Comment 4. You have determined parameter A to be 66.93. Putting it into equation on P6L171 as you did yield enormously high values for strain rate, about 1030. Please make sure that there is no mistake in your calculations or use of the formulae.
Response: Thanks for the Reviewer’s kind suggestion. The use of the formulae is right. According to the formulae, the calculated peak stress was shown in Figure 6. The results show that the correlation coefficient is 0.951(R > 0.95) and the average relative error is 4.26% (AARE < 5%). It indicated the constitutive equation has high fitting degree.
Comment 5. Figure 7 and P8L206. You have claimed on P8L206 that region with ξ<0 is rheological instability region; however, in figure 7 regions marked as instability region have obviously values of ξ>0 as follows from the isolines. Is there no mistake.
Response: Thanks for the Reviewer’s kind suggestion. The gray region ( < 0) is the rheological instability diagram (as shown below). And the isoline is the value of ln. They are two different quantities. So there are no mistakes.
Instability zones
Comment 6. Figure 7 again. Text below the figure says that the processing map is for fixed strain rate 0.5 s-1, but the figure itself shows a range of strain rates (vertical axis).
Response: Thanks for the Reviewer’s kind suggestion. We are really sorry for this kind of mistake. The processing map is for fixed strain, not the strain rate. So the unit should be deleted.
Comment 7. Variable N in text is lowercase, while in equation (15) it is uppercase. Is it the same variable? If it is, please make sure to use the same case in both instances.
Response: Thanks for the Reviewer’s kind suggestion. We are really sorry for this kind of mistake. N(Line 182) is the number of all data points. While in Line 266 and equation (15), it means the dislocation accumulation at the particle. The dislocation accumulation at the particle expressed in D. So, "n" should be changed to "D"(Line 266). The equation (15) has been modified to D.
In all, we found the reviewer’s comments are quite helpful and valuable in improving the quality of our manuscript, and we revised our paper point-by-point. Thank you and the reviewer again for your help!
Round 2
Reviewer 1 Report
There has been no in-depth review of the work. Only the mechanical results of the composite material with dual reinforcement versus a material with nanometric reinforcement have been contextualized. However, the dual reinforcement don´t represent any improvement over the nano-reinforcement. No justification is given to explain why the dual-reinforced material does not exhibit better properties than the nano-reinforced material. It is only said that the effects of both kind of particles in DRX is similar, although no evidence is given. In general, the requested improvements needed to publish the article have not been made. In particular, the first point referred to by the reviewer has not been answered. The authors say that this will be done in a future work. For all these reasons, my recommendation is to reject the paper.
Author Response
Manuscript number: Materials-821092
MS Type: Article
Title: Hot deformation behavior and microstructure evolution of the dual-scale SiCp/A356 composites based on optimal hot processing parameters
Correspondence Author: Aiqin Wang
Dear Editor and Reviewer,
Thank you for providing this great chance to revise the manuscript. We read the comments but didn't comprehend its full meaning. We feel terribly sorry for misunderstanding the reviewer's comments. We tried our best to improve the manuscript, and have revised the manuscript according to reviewer’s suggestions. Revised portion are made in red in paper. Enclosed please find the responses to the reviewers. There would be some controversy. And we think this is how science progresses. We sincerely hope this manuscript will be finally acceptable to be published on Materials. Thank you very much for all your help and looking forward to hearing from you soon.
With best regards,
Yours sincerely
Corresponding Author: Aiqin Wang
Response to the first point referred to by the reviewer. After we brush up the study on the high temperature creep behavior (References mentioned by the reviewer) and hot deformation behavior of aluminum alloys and its composites, they can all be regarded as a process of thermal activation energy. But the strain rate of hot deformation is several orders higher than that of creep. Moreover, in the study of the creep process of discontinuously reinforced aluminum (DRA) composites, load transfer mechanism and threshold stress (σo) have been introduced to explain the high creep strength and high stress exponent (n) and activation energy (Q) values of particle reinforced composites. As for the hot deformation behavior of DRA, only Cavaliere P mentions the flow stress behavior of DRA is government by two main processes: The transfer of load from the ductile matrix to the hard paritcles and the microstructural transformations such as DRX or damage phenomena; in this case the materials can present decohesion at the interfaces matrix-particles or several particle cracking. When the material is able to dissipate the provide power through the load transfer or through metallurgical transformations it does not reach high levels of damage. but the effect of the load transfer effect both on n (stress exponent) and Q (activation energy) isn’t described in detail in the flow stress behavior of DRA (Cavaliere P, Cerri E , Leo P. Hot deformation and processing maps of a particulate reinforced 2618/Al2O3/20p metal matrix composite[J]. Compos. Sci. Technol., 2004,64(9):1287 -1291 ).A modified equation including the load transfer effect to the hot deformation behavior of the DRA has not been reported.
Constitutive equations are of great importance in modeling the hot deformation processes, such as rolling, forging and extrusion. Over the past decades, various models have been proposed to describe the relationship between stress and strain rate for the high temperature stead state deformation. Garofalo has suggested a famous empirical hyperbolic-sine model to cover the dependence of steady state creep rate on stress at constant temperatures for both high and low stresses. Subsequently,on the basis of Garofalo’s equation, Sellars and Tegart have introduced another empirical hyperbolic-sine relationship (Arrhenius-type equations) to correlate the stress, strain rate and temperature under hot working conditions:
Where A is material constant,α is stress multiplier , n is stress exponent, Z is Zener–Hollomon parameter and Q is the activation energy. By its universality, it is the most widely used in the research of creep, superplastic and hot deformation behaviors of discontinuously reinforced aluminum (DRA) composite (Nardone V C, Strife J R. Analysis of the creep behavior of silicon carbide whisker reinforced 2124 Al(T4) [J]. Metall. Trans., 1987,18A: 109. Nieh T G, Xia K, Langdon T G. Mechanical properties of discontinuous SiC reinforced aluminum composites at elevated temperatures [J]. J. Eng. Mater. Technol., 1988, 110: 77. McQueen H J, Ryan N D. Constitutive analysis in hot working [J].Mater. Sci. Eng., 2002, A322:43).
As for the hot deformation behavior of metallic materials and AMCs, the relationship
between flow stress and strain rate and deformation temperature can be described by the Arrhenius-type equations (Sellars C.M., McG. Tegart W.J., On the mechanism of hot deformation, Acta Metall. 14 (1966)1136-1138. Sellars C.M., McG. Tegart W.J., Hot workability, Int. Metall. Rev. 17 (1972) 1-24. ). So, Arrhenius-type equations used in our research to explain the hot deformation of dual-scale SiCp/A356 composites are feasible. The calculated peak stress value by constitutive equation is compared with the tested value. The results show that the correlation coefficient is 0.951(R > 0.95) and the average relative error is 4.26% (AARE < 5%). It indicated the constitutive equation has high fitting degree and applicability. Although it is difficult for the constitutive equation to predict the deformation mechanism of DRA composites, the constitutive equation is the first basis to reveal the complex rheological behavior of DRA composites. Therefore, the establishment of perfect and accurate constitutive equation is the basis for promoting the development of processing technology of DRA composites. As the reference the reviewer mentioned, the modified equation including the load transfer effect provided a much more detailed description of the creep behavior might be applied to the detailed description of the hot deformation behavior of the DRA composites. This part is added to the manuscript(p7 L183-191). There would be some controversy. And I think this is how science progresses.
Response to the second point referred to by the reviewer. In general, the hot deformation activation energy Q is an important physical parameter that qualitatively reflects the energy barrier for dislocation motion during hot deformation.
The average Q value for the dual-scale SiCp/A356 composites is found to be 442.03kJ/mol. While the average activation energy for the nano-SiCp/Al-Si composites was 277 kJ/mol, which was larger than the activation energy for self-diffusion of pure aluminum (144 kJ/mol). The high Q value for nano-sized SiCp/Al-Si composites can be attributed to the nano-sized SiC hindering dislocations and the growth of grain boundaries. (Wang, Z.; Wang, A.Q.; Xie, J.P.; Liu, P. Hot deformation behavior and strain-compensated constitutive equation of nano-sized SiC particle reinforced Al-Si matrix composites. Materials. 2020, 13, 1812.). It is worthy to mention that the Q value of the dual-scale SiCp/A356 composites is much higher than the value for 15vol.% SiC/A356 (263 kJ/mol) and A356(161 kJ/mol) (McQueen H.J., Myshlyaev M., Konopleva E., Sakaris P., High temperature mechanical and microstructural behavior of A356/15 vol% SiCp and A356 alloy, Can. Metall. Q. 37 (1998)125-139). Chen X.R et al found that the Q value of Al-7 wt.% Si alloy reinforced by 10 wt.% micro-sized SiCp and 5 wt.%nano-sized TiB2 is 269.7 kJ/mol, which is much higher than the value for Al-Si alloy. The higher value of activation energy can be attributed to the addition of TiB2 reinforcements as a result of their serious retardation of dislocation movement in hot deformation. (Chen X.R, Fu D.F, Teng J., Hot deformation behavior and mechanism of hybrid aluminum-matrix composites reinforced with micro-SiC and nano-TiB2[J]. J. Alloy. Compd., 2018, 753:566-575.). It should be mentioned that the size of hard particles, the inter-particle spacing and their morphologies can affect the conditions for the occurrence of DRX. Irregular SiC particles with a large size are prone to breakage during deformation, which leads to a rotation of the broken particles and a weak interface between the particles and the Al matrix, and relaxes the stress concentration; while, the presence of TiB2 nanoparticles leads to typical strain hardening due to their pinning effect on dislocations. Besides, the Q value for nano-SiCp/Al-Si composites was slightly larger than that for Al-Si composites reinforced by micro-SiC, which may be attributed to the nano-SiC retarding the occurrence of the softening mechanism by hindering the growth of grain boundaries or subgrains. The difference in size between micro and nano particles was described in detail in the manuscript(P11 L271-277 and P12 L311-331 ).
Figure 1 shows the dual-scale SiCp/A356 with periodic cracking is not easy to undergo plastic deformation. It is consistent with the dual-scale SiCp/A356 composites show remarkably high values of Q. Figure 2 shows the Effects of content and size of SiC particles on the mechanical properties of the compacts. With the same volume fraction of SiC, the enhancement effect of dual-scale SiCp/A356 was more significant than that of single micron SiCp.
Figure 1. The hot extrusion bar: (a)A356,(b) 2% nano-sized SiC/A356 ,(c) 15% micro-sized SiC/A356,(d)dual-scale SiCp/A356
Figure 2. Effects of content and size of SiC particles on the mechanical properties of the compacts
In all, we found the reviewer’s comments are quite helpful and valuable in improving the quality of our manuscript, and we revised our paper point-by-point. Thank you and the reviewer again for your help!

Reviewer 2 Report
The manuscript may be accepted in the present form
Author Response
Dear Editor and Reviewer,
Thank you very much for all your help and looking forward to hearing from you soon.
With best regards,
Yours sincerely
Corresponding Author: Aiqin Wang